# An Overview on Renal and Central Regulation of Blood Pressure by Neuropeptide FF and Its Receptors

**DOI:** 10.3390/ijms252413284

**Published:** 2024-12-11

**Authors:** Hewang Lee, Jun B. Feranil, Pedro A. Jose

**Affiliations:** 1Department of Medicine, The George Washington University School of Medicine & Health Sciences, Washington, DC 20052, USA; jun.feranil@email.gwu.edu (J.B.F.); pjose01@gwu.edu (P.A.J.); 2Department of Pharmacology & Physiology, The George Washington University School of Medicine & Health Sciences, Washington, DC 20052, USA

**Keywords:** neuropeptide FF (NPFF), NPFF receptor 1 (NPFFR1), NPFF receptor 2 (NPFFR2), hypothalamus, paraventricular nucleus, brainstem, kidney, blood pressure

## Abstract

Neuropeptide FF (NPFF) is an endogenous octapeptide that was originally isolated from the bovine brain. It belongs to the RFamide family of peptides that has a wide range of physiological functions and pathophysiological effects. NPFF and its receptors, NPFFR1 and NPFFR2, abundantly expressed in rodent and human brains, participate in cardiovascular regulation. However, the expressions of NPFF and its receptors are not restricted within the central nervous system but are also found in peripheral organs, including the kidneys. Both NPFFR1 and NPFFR2 mainly couple to Gαi/o, which inhibits cyclic adenosine monophosphate (cAMP) production. NPFF also weakly binds to other RFamide receptors and the Mas receptor. Relevant published articles were searched in PubMed, Google Scholar, Web of Science, and Scopus. Herein, we review evidence for the role of NPFF in the regulation of blood pressure, in the central nervous system, particularly within the hypothalamic paraventricular nucleus and the brainstem, and the kidneys. NPFF is a potential target in the treatment of hypertension.

## 1. Introduction

The pathogenesis of hypertension involves a complex interplay among behavior, genetics, microbiome, and environment [1], involving multiple organs, including the peripheral and central nervous system (CNS), vascular system, and endocrine system [2,3,4]. However, the kidney has the primary role in setting the blood pressure level in the long term [2,5,6]. A deviation from the blood pressure set point causes a proportional change in renal arterial perfusion pressure, resulting in a parallel change in sodium excretion, which subsequently changes the circulatory volume and cardiac output [2,5,6]. Central to this process, in which the kidney has the overriding dominance, is pressure natriuresis in which the increase in arterial blood pressure is sensed by the kidney, leading to a decrease in sodium reabsorption along the nephron, increasing sodium excretion [2,5,6]. The brain also receives information on the alterations of blood pressure via sensory inputs from peripheral organs, including the kidneys, and organizes a response by the sympathetic and parasympathetic autonomic nervous systems and the release of hormones to restore homeostasis [3,7,8,9]. The dysregulation of vascular resistance in arteries and arterioles in hypertension [4,10] is not covered in this review. 

Genome-wide association studies (GWAS) have identified ~30 genes and ~1500 single nucleotide variants to be either causal or associated with hypertension [11,12]. However, the clinical translation of the GWAS data is challenging and has not yet led to a genetically based treatment of hypertension [13]. Thus, there is still a need to identify novel genes responsible for the pathogenesis of hypertension. One such gene is *Npff (NPFF)*, encoding the protein neuropeptide FF (NPFF), which has been implicated in the increase in blood pressure in animals and humans [14]. 

In this review, PubMed, Google Scholar, Web of Science, and Scopus were searched using the key words “neuropeptide FF”, “NPFF”, “neuropeptide”, and “RFamide peptide” with “hypertension”, “essential hypertension”, “blood pressure”, “NPFF receptor”, “GPCR”, “signaling”, “signal transduction”, “brain”, “central nervous system”, “CNS”, “kidney”, and “renal” in various combinations with no date limitation until April 2024. After exclusion of conference abstracts, a total of 1347 peer-reviewed articles were obtained and evaluated. Their relevance was initially assessed by removing articles with titles containing “pulmonary hypertension” or “portal hypertension”. The references from the relevant articles were also checked.

## 2. NPFF and NPFF Receptors

NPFF (FLFQPQRF-NH2), a neuropeptide originally isolated from bovine brain, is a pain-modulating peptide with anti-opioid activity in rats [15]. As shown in Table 1, the NPFF system has two peptide precursors, pro-NPFF_A_ and pro-NPFF_B,_ with two G protein-coupled receptors, NPFFR1 (GPR147, OT7TO22) and NPFFR2 (GPR74, HLWAR77) [16,17,18,19]. RFamide-related peptide (RFRP)-1 and RFRP-3 or neuropeptide VF (NPVF), derived from pro-NPFF_B_, have a higher affinity for NPFFR1 than NPFFR2; NPFF, derived from pro-NPFF_A_, has a higher affinity for NPFFR2 than NPFFR1 [20].

NPFF belongs to the family of RFamide peptides that has a common carboxy-terminal arginine (R) and an amidated phenylalanine (F) motif [21,22]. The RFamide family, in mammals, has five members: (1) NPFF (PQRF amide) group [NPFF and neuropeptide AF (NPAF)] [21,22]; (2) gonadotropin inhibitory hormone (GnIH) (RFamide-related peptide group [RFRP1 and RFRP3]) [21,22,23]; (3) pyroglutamylated RF amide peptide [QRFP]) group [43RFa and 26RFa] [21,22,24]; (4) kisspeptin group [e.g., kisspeptin-10,-13, -14, -54] [21,22,25]; and (5) prolactin-releasing peptide (PrRP) group [PrRP31 and PrRP20] [21,22,26]. These five groups of RFamide peptides have their own cognate receptors: NPFFR1 (GPR147) for RFRP1, RFRP3, and NPFF; NPFFR2 (GPR74) for NPFF and NPAF; QRFPR (GPR103) for QRFP, KISS1R (AXOR12, GPR54) for kisspeptins; and PRLHR (GPR10, GR3) for prolactin-releasing peptides [21,22]. The mammalian RFamide family of peptides are summarized in Table 2 [14,21,22,23,24,25,26].

NPFF is widely expressed in the brain and spinal cord [27,28,29,30,31,32,33]. Early Northern blots showed that NPFF is expressed in the rat hypothalamus and brainstem [27], areas involved in cardiovascular regulation. In situ hybridization using radioactive labeling showed that NPFF is robustly expressed in the nucleus tractus solitarius (NTS), spinal trigeminal nucleus (SpV), paraventricular nucleus (PVN), and supraoptic nucleus (SON) [27], which was confirmed by RNAscope that NPFF is primarily expressed in the caudal brainstem, including SpV, and the posterior residual brain region and spinal cord [28,29]. In the brainstem, NPFF neurons and fibers are abundant in the dorsal vagal complex [30,31]. Many NPFF fibers form synaptic-like contacts with neuronal profiles in the dorso-, centro-, ventro-, and caudo-intermediate of the dorsal motor nucleus of the vagus nerve, and medial and intermediate subnuclei of the nucleus tractus solitarius [30,31]. The expression of NPFF in the brain is summarized in Figure 1.

Both NPFFR1 and NPFFR2 are widely but differentially expressed in the CNS [16,17,18,19,20,33,34,35,36], indicating distinct roles in mediating the actions of NPFF and its related peptides. NPFFR1, identified by autoradiography, is moderately expressed in the hypothalamus and dorsal and intermediate lateral septal nucleus in mice [33,34,35] and rats [16,33,34,35]. NPFFR1 is minimally expressed in the spinal cord and mesencephalon, except in the superior colliculus in mice and rats [34]. In situ hybridization with isotype oligonucleotides showed that NPFFR1 is also expressed in the brainstem and limbic system, including the stria terminalis, pars compacta of the substantia nigra, periaqueductal gray, dorsal raphe, lateral parabrachial nucleus, locus coeruleus, and the dorsal motor nucleus of the vagus [19,34]. RNA in situ hybridization also showed that NPFFR1 is expressed in regions related to neuroendocrine activity, including the PVN, arcuate nucleus (ARC), and brainstem, such as the periaqueductal gray, dorsal raphe, and nucleus sagulum [35]. Our immunohistochemical staining demonstrated that NPFFR1 is expressed in multiple hypothalamic nuclei, including the PVN, arcuate nucleus, and ventromedial hypothalamus, but is minimally expressed in the dorsomedial hypothalamus [36].

The NPFFR2 protein is widely and strongly expressed in hypothalamic nuclei, including the organum vasculosum of the lamina terminalis, supraoptic nucleus, PVN, arcuate nucleus, and ventromedial hypothalamus, and in non-hypothalamic regions, such as the hippocampus, cerebral cortex, and piriform cortex in the mouse brain, determined by immunohistochemistry [36]. Autoradiographic studies, using specific radiolabeled ligands, also showed that NPFFR2 is highly expressed in the nucleus accumbens, thalamus, hypothalamus, and the suprachiasmatic nucleus [16,17,18,19,34]. In the brainstem, reverse transcription polymerase chain reaction (RT-PCR) and autoradiographic studies showed that NPFFR2 is expressed in the dorsolateral periaqueductal gray, cuneiform nucleus, dorsal raphe nucleus, dorsal tegmental nucleus area, dorsal cortex of the inferior colliculus, interpeduncular nucleus, medial mesencephalic geniculate nucleus, parabigeminal nucleus, pontine reticular nucleus, peripeduncular nucleus, superior colliculus, and thalamic submedius nucleus [16,17,18,19]. RNA in situ hybridization also showed that *Npffr2* mRNA is expressed in the subnuclei of the thalamus, glossopharyngeal, and vagus nerves exiting the medulla oblongata, posterior lateral sulcus, lateral lemniscus, and nucleus tractus solitarius [35]. The wide expression of NPFFR1 and NPFFR2 in the CNS [16,17,18,19,33,34,35,36] suggests that the NPFF system has multiple functions, including the regulation of cardiovascular function.

**Figure 1 ijms-25-13284-f001:**
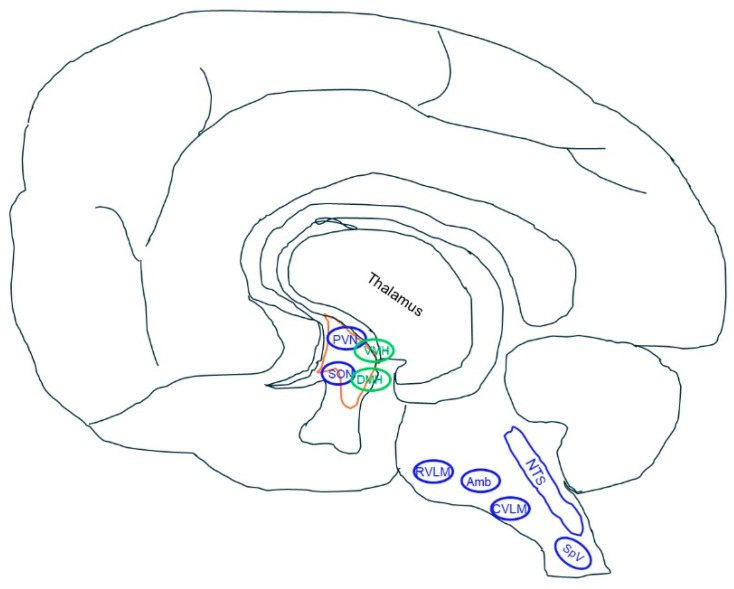
Schematic diagram of NPFF expression associated with blood pressure regulation in the central nervous system (CNS). In situ hybridization shows that NPFF is mainly expressed in mouse caudal brainstem and spinal cord [28], presumably including the NTS, RVLM, CVLM, and nucleus Amb (in blue); additional expression is also reported in PVN, SON, VMH, and DMH using in situ hybridization [27] and immunohistochemistry (in green) [30]. NPFF is highly expressed in the spinal cord (mainly in the dorsal horn) in rats, mice, and humans [27,28,29,30]. In mice, NPFFR1 is expressed in multiple hypothalamic nuclei, including the PVN, SON, ARC, OVLT, and VMH and non-hypothalamic regions, such as HP and CTX. NPFFR2 is widely and highly expressed in hypothalamic nuclei, including the OVLT, SON, PVN, ARC, and VMH, and in non-hypothalamic regions such as the HP, Pir, and CTX [36]. In humans, NPFF is also expressed in the gray matter of the frontal, cingulate, superior temporal gyri, and superficial white matter in human brains [32]. NPFF and its receptors play an important role in the regulation of blood pressure, presumably involving baroreflex, sympathetic, parasympathetic (vagal), and neuroendocrine mechanisms. Amb, nucleus ambiguus; ARC, arcuate nucleus; CTX, cerebral cortex; CVLM, caudal ventrolateral medulla; DMH, dorsomedial hypothalamus; HP, hippocampus; NTS, nucleus tractus solitarius; OVLT, organum vasculosum of the lamina terminalis; Pir, piriform cortex; PVN, paraventricular nucleus; RVLM, rostroventrolateral medulla; SON, supraoptic nucleus; SpV, spinal trigeminal nucleus; VMH, ventromedial hypothalamus. Of note, the diagram is depicted in human brain, assuming NPFF expression is highly conserved in rodents and humans.

The expression and distribution of NPFFR1 and NPFFR2 in peripheral tissues, especially in the kidney, are not fully known. Limited mRNA expression of *Npff* is observed in the adrenal gland, heart, lung, pancreas, skin, and spleen [16,17,18,19,33,34,37]. However, the adipose tissue and pancreas have considerable *Npff* expression in mice [38], consistent with its role in the regulation of glucose and lipid metabolism [38,39]. There is minimal *Npffr1* mRNA expression in the adrenal gland, eye, intestine, kidney, lung, ovary, and spleen [16,17,18,19,30]. By contrast, *Npffr2* mRNA is highly expressed in adipose tissue, heart, kidney, retina, salivary gland, stomach, and urinary bladder [16,17,18,19,33,36]. Recent single-cell RNA sequencing showed considerable *NPFFR1* and detectable *NPFFR2* expression in human renal proximal tubule cells (RPTCs) (https://www.proteinatlas.org/humanproteome/tissue+cell+type, accessed on 1 October 2024). However, considerable discrepancies in NPFF receptor expression exist in studies in the CNS and peripheral tissues that may be species-related and use different methods [16,17,18,19,20,27,28,29,30,31,32,33,34,37,40,41].

## 3. NPFF Receptors and Their Signaling 

NPFFR1 and NPFFR2, as with all seven transmembrane spanning receptors, act via heterotrimeric guanine nucleotide regulatory proteins (G proteins). Initial studies showed the ability of guanine nucleotides to inhibit the binding of NPFF to its receptors, suggesting that both NPFFR1 and NPFFR2 receptors are coupled to G proteins [16,17,18,19]. Both NPFFR1 and NPFFR2 dose-dependently decrease the forskolin-stimulated increase in cyclic adenosine monophosphate (cAMP) production in Chinese hamster ovary (CHO) cells transfected with human NPFFR1 and NPFFR2, suggesting that NPFFR1 and NPFFR2 are primarily coupled to the inhibitory Gα subunit, Gαi/o protein, and inhibit adenylate cyclase activity [42]. However, NPFFR2 can also couple to Gαs and stimulate adenylate cyclase activity in the mouse olfactory bulb and rat spinal cord membranes [43], suggesting the possibility of a Gαi/o-Gαs switch of cAMP signaling. 

NPFF does not bind to opioid receptors [44]. However, NPFF is an endogenous opioid modulating peptide with different nociception effects, depending on the site of administration, where NPFFR1 and NPFFR2 are distinctly expressed [15,16,17,18,19,33,34,41]. The pronociceptive effects of intrathecal administration of NPFF analogs are most likely through NPFFR2 activation, whereas the antinociceptive effects of intracerebroventricular administration are mediated mainly by NPFFR1 [41]. However, the underlying signaling pathways of NPFF receptors on their nociceptive effects are not known.

In HEK293 [16], COS-7 [16], CHO [18], and SH-SY5Y [45] cells, NPFFR1 is mainly coupled to Gαi and inhibition of cAMP production. Presumably, its antinociceptive and anti-opioid effects are through inhibition of Ca^2+^ signaling [41]. In NPFFR1-transfected SH-SY5Y cells, NPVF (NPFFR1 agonist) reduces the depolarization-induced calcium transients in a concentration-dependent manner, and activation of NPFFR1 by NPVF reduces the inhibitory effect of opioid on polarization-induced [Ca^2+^]_i_ transients, indicating that NPVF exerts a functional antagonism of opioid receptors [45]. In gonadotropin-releasing hormone (GnRH) neurons, NPFF activates NPFFR1 and inhibits GnRH neuron excitability via the canonical Gαi/o protein and G protein-coupled inwardly rectifying potassium channel [46], which are independent of GABAergic and glutamatergic activities [46].

Similar to NPFFR1 [40,45], NPFFR2 is predominantly coupled to the Gαi/o subunit to inhibit cAMP production [17,19,42] and N-type voltage-gated calcium channels [47]. NPFFR2 can also be linked to Gαs, as well as activation of G protein-regulated inwardly rectifying potassium channels in pain regulation [48] and a putative delayed rectifier K channel in dorsal root ganglion neuron cells [49]. In SH-SY5Y cells, NPFFR2 activates extracellular signal-regulated kinase (ERK)1/2 mitogen-activated protein kinase pathway [50,51], which is pertussis toxin-sensitive [50]. Two loci of agonist-induced phosphorylation within NPFFR2 have been identified to be involved in NPFFR2-mediated desensitization and/or internalization using combined methods of mass spectrometry, mutants, and anti-phospho-threonine antibodies [50]. However, the kinase(s) responsible for its phosphorylation have not been identified. Mutation of aspartic acid (D) at position 3.49 with alanine (A) from the highly conserved DRF motif in the second intracellular loop does not impair the binding capacities of receptor-specific ligands, whereas this amino acid is essential for NPFFR2-mediated signal transduction [52]. Truncation of the NPFFR2 C-terminus showed that the putative eighth α-helix and acylation site is essential for its ligand binding, correct folding, and signal transduction [52].

In addition to NPFFR1 and NPFFR2, NPFF can activate neuropeptide Y (NPY) receptors with 30–35% homology [16,20]. BIBP3226 binds to NPFF and NPY1 receptors, and the frog pancreatic polypeptide, a NPY4 agonist, exhibits nanomolar affinity for NPFFR2 [16,42]. In human hypothalamic ARC-like neurons, NPFF decreases neuronal activity, as shown by the reduction of calcium spikes, a well-described electrophysiological characteristic of arcuate NPY neurons [53].

NPFF is also linked to atypical signaling by activation of the Mas receptor (MasR), a G protein-coupled receptor encoded by the proto-oncogene MAS that also plays an important role in blood pressure regulation [54]. NPFF, via MasR, causes a rapid rise in intracellular calcium with minimal increase in the accumulation of D-myo-inositol-1-phosphate, unlike the classical Gαq-phospholipase C signaling pathway [16,54], which is distinct from its effect of a reduced rate of cytoplasmic calcium oscillations through NPY in hypothalamic ARC-like neurons [53], indicating different effects of NPFF in different cells. The MasR-mediated activation of the Gαi and Gα_12_ signaling pathways is weakly stimulated by NPFF [55].

In human and mouse RPTCs, either NPFFR1 or NPFFR2 inhibits forskolin-stimulated cAMP production in a concentration- and time-dependent manner [36], which is reversed by pretreatment with RF-9, a well-known NPFFR1 and NPFFR2 antagonist [56]. This indicates that both NPFFR1 and NPFFR2 are mainly coupled to Gαi/o to inhibit adenylyl cyclase activity in the kidney, similar to its signaling in neurons and other cells (e.g., transfected HEK293 cells, CHO cells, and SH-SY5Y neuroblastoma cells) [17,20,41,42,44,47]. In COS-7 cells transfected with NPFFR1 or NPFFR2, NPFF and its analogs inhibit forskolin-stimulated cAMP production [16]. However, as aforementioned, the NPFF analog can activate adenylyl cyclase and increase cAMP production in the membranes of the mouse olfactory bulb [43] and rat spinal cord [44], where NPFFR2 is expressed [33,41]. Therefore, NPFFR2 can also couple to Gαs. It remains to be determined whether NPFF activates atypical signaling and NPY signaling in their regulation of renal sodium transport and, subsequently, blood pressure.

NPFF receptors, like most G protein-coupled receptors, have multiple signaling pathways resulting from their G protein couplings that account for their versatile cellular actions (Figure 2). Further studies are needed to elucidate the particular signaling network in central and peripheral cells that are altered in hypertension.

## 4. Role of NPFF and Its Receptors in the Regulation of Blood Pressure in the Central Nervous System

NPFF can regulate diverse physiological functions, including adipose metabolism, body temperature, food intake, gastrointestinal motility, immune activity, nerve injury repair, and water balance, among others [28,33,41]. The distribution of NPFF and its receptors in the cardiovascular regulatory centers in the hypothalamus and the brainstem indicates their role in the regulation of blood pressure [14,30,31,32,33,34,35]. Blood pressure is increased by the intraventricular injection of the FMRFamide peptide [57]. The intracerebroventricular [57,58,59], intra-nucleus tractus solitarius [60], and intrathecal [61] administration of NPFF increases blood pressure, proving that NPFF and its receptors in the central nervous system positively regulate blood pressure. 

The intracerebroventricular administration of NPFF in rats induces NPFF gene expression in the nucleus tractus solitarius [58] and increases arterial blood pressure [58,59]. The increase in arterial blood pressure occurs within 1 min and is sustained for 25 min before returning to normal levels. The NPFF-mediated increase in blood pressure is blocked by RF9 [59], a selective NPFF receptor antagonist [56], indicating that the NPFF-mediated increase in blood pressure is through its receptors. The intracerebroventricular administration of NPFF also prevents the decrease in blood pressure caused by intravenous injection of the µ-opioid receptor agonist endomorphin-1, effects that are prevented by the simultaneous intracerebroventricular injection of NPFF and RF9 [59]. Bilateral microinjection of NPFF into the commissural nucleus tractus solitarius in rats also increases blood pressure [60]. In anesthetized rats, the intrathecal administration of NPFF, NPVF (NPFFR1 agonist), or dNPA (NPFFR2 agonist) increases blood pressure, which is prevented by RF9, indicating that the increase in blood pressure by NPFF, NPVF, and dNPA is mediated by both NPFFR1 and NPFFR2 in the spinal cord [61]. All these reports demonstrate the role of NPFF in the central nervous system, including the hypothalamus, brainstem, and spinal cord, in the regulation of blood pressure.

Prolactin-releasing peptide (PrRP) binds with high affinity to NPFF receptors, specifically NPFFR2 [62]. In conscious rats, the intracerebroventricular administration of PrRP increases arterial blood pressure, which is blocked by the NPFFR1 and NPFFR2 antagonist RF9 [63], suggesting that the PrRP-evoked blood pressure increase is most likely mediated by NPFFR2. 

The mechanisms by which NPFF increases blood pressure are not well known. The adrenergic system may be involved, because the heart rate is increased with the increase in blood pressure caused by the intracerebroventricular [58,59,60], intrathecal [61], and intravenous [64] administration of NPFF. The pressor and tachycardic effects of NPFF are blocked by α_1_-adrenoceptor antagonists [60,61,64]. The high expression of NPFF and its receptors in the hypothalamus and medulla suggests that neuroendocrine and cardiovascular centers are involved in the NPFF regulation of blood pressure [14,33,34,35]. Two major neurosecretory cells, magnocellular and parvocellular, are in the PVN [14]. The hypothalamus–pituitary–adrenal axis regulates blood pressure via the autonomic and neuroendocrine systems [65,66,67,68]. Vasopressin and oxytocin are the two major hormones produced in the PVN [14,69] and stored and released by the posterior pituitary gland [14]. RFRP-3, a NPFFR1 agonist, attenuates the morphine-mediated inhibition of the spontaneous firing rate of vasopressin and oxytocin neurons [70]. In rats, *Npffr1* is expressed in the lateral septal nucleus and several hypothalamic areas, including corticotropin-releasing, tyrosine hydroxylase-positive, and dopaminergic neurons in the PVN and arcuate nucleus, but minimally expressed in vasopressin and oxytocin neurons [35]. *Npffr2* is mainly expressed in neurons and fibers in hypothalamus, thalamus, and brainstem involved in multiple neuroendocrine pathways [33,34,35,36,58], presumably by activating the hypothalamic–pituitary–adrenal (HPA) axis. Many hormones in the anterior pituitary gland, such as the corticotropin-releasing hormone, directly or indirectly, participate in the regulation of blood pressure [65,67,68,71]. NPFF differentially regulates GABAergic input to the parvocellular and magnocellular neurons of the PVN [14]. NPFF inhibits the activity of GABAergic terminals that project to the parvocellular PVN neurons, resulting in their “disinhibition”, whereas it augments GABA synaptic input to magnocellular PVN neurons, resulting in their inhibition [14]. The inhibition of GABAergic activity enhances the release of the corticotropin-releasing hormone, leading to an increase in blood pressure [14,72]. NPFF expression is decreased in the hypothalamus of hypertensive patients, relative to matched healthy controls with systolic blood pressures less than 140 mm Hg [73]. In the PVN parvocellular division, GABAergic neurons strongly colocalize with G_αi2_ [69], which is coupled with NPFFR2 [41,74], indicating that NPFFR2 can potentially regulate salt sensitivity and blood pressure through GABAergic activity. In mice, the NPFFR2 agonists dNPA and AC-263093 dose-dependently increase circulating corticosteroid concentrations [75], which are prevented by RF9 (NPFF receptor antagonist) and α-helical CRF (CRF antagonist) [75]. Thus, NPFFR2 can activate the HPA axis in rodents. The HPA axis is altered in hypertension [65,66,67,68]. 

In the brain, most excitatory signals are mediated by glutamate receptors (e.g., NMDA receptor), whereas most inhibitory signals are mediated by GABA receptors [76]. Proper brain function requires the balance of excitatory and inhibitory signals [76]. GABAergic receptors regulate the expression, activity, and signaling of NMDA receptors under physiological and pathological conditions and vice versa [77]. As discussed above, the NPFF system regulates GABAergic activity [14,73]. Therefore, the NPFF system can potentially affect NMDA activity directly or indirectly through GABA receptors, but the mechanisms by which NPFF regulates NMDA activity in the brain are not known.

In the nucleus tractus solitarius, NPFF receptors are preferentially located postsynaptically in vagal afferent fibers [60]. The microinjection of NPFF into the nucleus tractus solitarius increases blood pressure, decreases heart rate, and inhibits the cardiac component of the baroreceptor reflex [60]. This is consistent with the observation that baroreflex sensitivity, mainly maintained by autonomic brainstem nuclei [78], is decreased in hypertensive animals and humans [79]. In normotensive Wistar Kyoto rats, the sympathetic activity in the perifornical and lateral hypothalamic areas decreases as the normotensive rats grow older, whereas such decrease is not observed in spontaneously hypertensive rats [80]. NPFF expression also decreases with age in the nucleus tractus solitarius and dorsal motor nucleus of the vagus, the main nerve of the parasympathetic nervous system [80]. The increase in sympathetic activity and decrease in parasympathetic activity in the central nervous system are associated with the development of hypertension [80].

NPFF concentrations in the septum and hippocampus are low but with moderate densities of NPFF-containing neurofibers [28,33]. NPFFR2 is strongly expressed in the hippocampus, amygdala, and anterior nuclei of the thalamus. NPFFR1 is also expressed in the hippocampus and amygdala [34]. Neuropeptide B/W receptor 1 is abundantly expressed in the hippocampus and central nucleus of the amygdala [81]. Mice with a deficiency of neuropeptide B/W receptor 1 have increased blood pressure relative to their wild-type counterparts [81]. However, the role of NPFF and its receptors in the regulation of blood pressure by the limbic system is unknown. 

The injection of NPFF into spinal cord increases blood pressure, but the mechanism is not well known [61]. In rats, the increase in blood pressure caused by the spinal cord administration of NPFF is partially decreased by the α_1_ and α_2_-adrenergic antagonist phentolamine, suggesting a possible role of α-adrenoceptors in the hypertensinogenic effect of NPFF [61]. Glutamate may also be involved, because it can increase the release of NPFF from the rat spinal cord, which is blocked by NMDA receptor antagonists, suggesting that NMDA receptors are also involved in the NPFF-mediated increase in blood pressure via the spinal cord [82].

Anatomical, physiological, and pharmacological studies suggest that NPFF in the central nervous system plays an important role in neuroendocrine regulation [14], involving the baroreceptor reflex [60], and GABAergic [14,73], muscarinic [45,61,83], α and β adrenergic [61,64,83], serotoninergic [83], dopaminergic [35,83], and NMDAergic [83,84] systems (Figure 3). 

## 5. Roles of NPFF and Its Receptors on the Regulation of Blood Pressure Outside the Central Nervous System

NPFF does not cross the blood–brain barrier [41,59,64] but can access the permeable areas involved in the control of sympathetic tone and regulation of blood pressure (e.g., circumventricular organs in the area postrema and nucleus tractus solitarius) [64]. NPFF and its receptors, NPFFR1 and NPFFR2, are also present outside the central nervous system, including the kidney [16,17,36,41]. The systemic administration of NPFF increases blood pressure in anesthetized [59,64,85] and conscious rats [86]. The NPFF-mediated increase in blood pressure is attenuated but not prevented by prazosin, an α_1_-adrenoreceptor antagonist [41,64,86], indicating the involvement of both adrenergic and non-adrenergic mechanisms. As aforementioned, the increase in blood pressure caused by the intrathecal administration of NPFF is decreased by the α_1_ and α_2_-adrenergic antagonist phentolamine [61]. The increase in blood pressure caused by NPFF is probably exerted at the receptor level, because the intravenous administration of NPFF induces a dose-dependent increase in blood pressure without affecting the plasma epinephrine and norepinephrine levels [64]. Indeed, a NPFF antagonist prevented the NPFF-induced elevation of blood pressure in anesthetized rats [36,56]. Moreover, in anesthetized rats, the intravenous injection of PFRFamide, a putative NPFFR1 and NPFFR2 agonist, dose-dependently increases the mean arterial blood pressure, whereas PFR(Tic)amide, a NPFFR1 antagonist, decreases blood pressure [41,87]. Therefore, these studies suggest the role of NPFF in the regulation of blood pressure outside the central nervous system.

NPFF and its receptors in the kidney may participate in the regulation of blood pressure. We found that the transcripts and proteins of NPFF and its receptors, NPFFR1 and NPFFR2, are expressed in the human and mouse RPT [36]. In mouse RPTCs, NPFF, but not RFRP-2, decreases the forskolin-stimulated cAMP production in a concentration- and time-dependent manner [36]. In the nucleus tractus solitarius, NPFF can bind to RFRP, which is co-expressed with neuropeptide Y (NPY) [46]. NPY and three of its receptors (Y1, Y2, and Y5) are present in RPTs, whose activation can increase blood pressure [88]. Dopamine D1-like receptors can inhibit the smooth vascular muscle proliferation caused by NPY [89]. The inhibition of vascular smooth muscle cell proliferation can be induced by cGMP-dependent activation of cAMP kinase [90]. Because D1-like receptors are transduced, in part, by cAMP [89], and NPFF can inhibit cAMP production [36], it is possible that NPY contributes to the NPFF-induced cAMP signaling and subsequent blood pressure regulation.

Studies from our laboratory [91,92,93,94,95,96,97] and others [98,99,100] have shown that the dopaminergic system in the kidneys is important in the regulation of normal blood pressure. We recently found that dopamine D1-like receptors colocalized and co-immunoprecipitated with NPFFR1 and NPFFR2 in human RPTCs [36]. The increase in cAMP production caused by fenoldopam, a D1-like receptor agonist, was attenuated by NPFF in mouse RPTCs [36]. Therefore, there is an antagonistic interaction between NPFF and D1-like receptors, at least in the kidney. To determine the renal-selective effect of NPFF on sodium excretion and blood pressure, we infused NPFF underneath the renal capsule of anesthetized mice [36]. We and others have reported that the renal interstitial or subcapsular infusion of drugs or oligonucleotides restricts their actions in the kidney independent of other organs [101,102,103,104]. The acute and chronic renal subcapsular infusion of NPFF in C57BL/6 mice decreased renal sodium excretion and increased arterial blood pressure [36]. The systolic blood pressure was increased 15–20 min after the renal subcapsular infusion of NPFF, which was prevented by RF9, a NPFFR1 and NPFFR2 antagonist, which by itself had no effect on blood pressure or sodium excretion. Scrambled peptides also had no effect on blood pressure or sodium excretion [36]. Although RF9 has been reported to be a selective antagonist of NPFFR1 and NPFFR2 [56], recent studies have shown that RF9 activates KISS1R (GPR54) [105,106,107] and subsequently GnRH neurons [106] and the release of luteinizing hormone [108]. Thus, the ability of RF9 to block the NPFF-mediated increase in blood pressure via the kidney [36] and CNS [56,59,63,75] could involve effects on other RFamide peptide receptors.

To determine the molecular mechanisms by which NPFF increases renal sodium transport, we measured the protein expressions of Na^+^/K^+^-ATPase and sodium-hydrogen exchanger 3 (NHE3) by immunoblotting the kidney cortices of normal salt diet-fed C57BL/6 mice with a deficiency in *Npffr1* or *Npffr2* induced by the renal subcapsular infusion of specific *Npffr1* and *Npffr2* siRNAs [36]. The renal protein expression of NHE3 was slightly decreased in mice with renal subcapsular infusion of *Npffr1* siRNA but was markedly increased in those infused with *Npffr2* siRNA. Renal phosphorylated NHE3 expression was not affected by *Npffr1* or *Npffr2* siRNA. Renal Na^+^/K^+^-ATPase protein expression was also minimally affected in those mice. Therefore, renal NHE3 and Na^+^/K^+^-ATPase protein expression cannot explain the ability of NPFF to increase renal sodium transport, decrease sodium excretion, and increase blood pressure. NPFFR2 is found in the choroid plexus epithelial cell cilia, and its activation decreases cellular cAMP levels and basolateral to apical fluid transcytosis [109]. Therefore, autocrine NPFF and its receptors in the kidney may regulate blood pressure by increasing renal tubular sodium transport, but the mechanisms remain to be determined (Figure 4). 

## 6. Conclusions and Perspectives

NPFF belongs to the RFamide family. NPFF in both the central nervous system and kidneys play important roles in the regulation of blood pressure and pathogenesis of hypertension. NPFFR1 and NPFFR2 are capable of binding to all RFamides with varying degrees of affinity and potency [110,111]. NPFF can also bind to other receptors, such MasR [55], Mas-related G-protein coupled receptors (Mrgprs) C11 [112], and Mrgprs A4 [113], which activate atypical signaling. Kisspeptins also bind and activate both NPFFR1 and NPFFR2, activate Gαi/o signaling, and inhibit cAMP production and voltage-gated Ca^2+^ channels [114]. The relationship of these downstream signaling events with the ability of NPFF and its receptors to increase renal sodium transport remains to be determined. The promiscuity between ligands and receptors within this family of RFamide peptides [110,111] limits the choice for specific agonists and antagonists. The generation of specific agonists and antagonists for the five subfamilies of RFamide peptide receptors will certainly advance our understanding of the role of these peptides and receptors, including NPFF, NPFFR1, and NPFFR2 in the blood pressure regulation. The technical advances in medicinal chemistry could lead to the generation of these specific pharmacological reagents [115].

Hypertension is a multifactorial disorder influenced by behavior, genetics, microbiome, and environment [1]. Genome-wide association and candidate gene studies have revealed various single-nucleotide polymorphisms (SNPs) associated with hypertension [11,12,13]. A SNP of *NPFF*, rs11170566, is associated with migraine headaches, inflammation, and cardiovascular disorders [116]. SNPs of *Npffr1* may be related to growth-related traits of the common carp [117]. A common haplotype of *NPFFR2* (ATAG) is associated with leanness and increased lipid mobilization [118]. *Npffr2* rs110326785 is associated with mastitis in dairy cattle [119]. The association of SNPs of NPFF and its receptors with inflammation and aberrant metabolism of glucose and lipids [118,119] and the ability of NPFF and its receptors to increase blood pressure are consistent with the notion that hypertension is a chronic inflammatory disorder [120,121]. Gene editing holds the potential to target and correct genetic mutations that contribute to the development of hypertension or modify the involved genes to prevent or treat this condition. Advances in computational biology and machine learning enable large data analyses, which could lead to the identification of complex genetic interactions that contribute to the pathogenesis of hypertension.

## Figures and Tables

**Figure 2 ijms-25-13284-f002:**
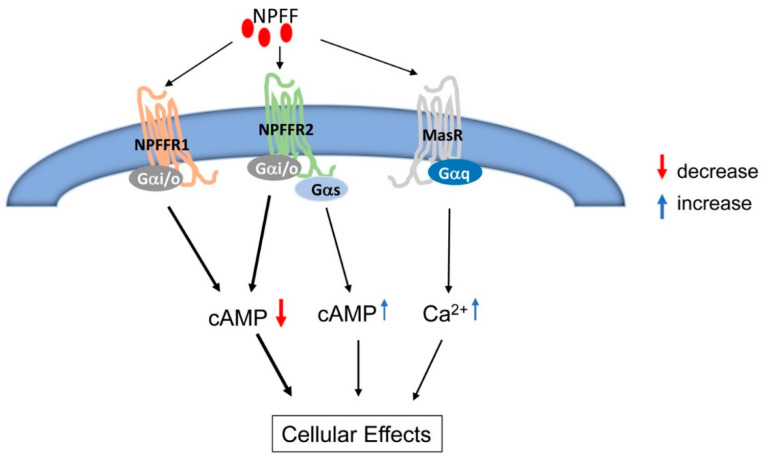
Schematic diagram of NPFF and its receptors’ signaling pathways. NPFF binds to NPFFR1, NPFFR2, MasR, and other RFamide receptors. NPFFR1 and NPFFR2 are primarily coupled to inhibitory Gαi/o and inhibit adenylate cyclase activity. NPFFR2 can also couple to Gαs and stimulate adenylate cyclase activity in the mouse cerebellum, olfactory bulb, and spinal cord. NPFF weakly binds to MasR, activating an atypical Gαq-phospholipase C signaling pathway. The thick arrows show the principal mechanisms of NPFFR1 and NPFFR2 activation, while the thin arrows show ancillary mechanisms of NPFFR2 activation.

**Figure 3 ijms-25-13284-f003:**
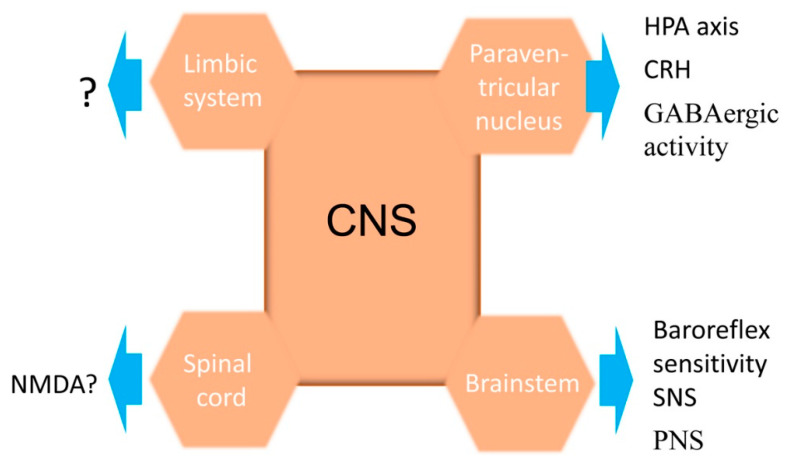
Hypothetical CNS-mediated regulation of blood pressure by NPFF. NPFF regulates blood pressure via its receptors NPFFR1 and NPFFR2 in the hypothalamus, the brainstem, limbic system, and spinal cord through the hypothalamus–pituitary–adrenal (HPA) axis, baroreceptor reflex, GABAergic, muscarinic, α- and β-adrenergic, serotoninergic, dopaminergic, and NMDAergic systems. The chronic increase in corticotropin-releasing hormone signaling associated with dysregulated signaling of the HPA axis increases the cortisol levels and, subsequently, blood pressure. In concert with the release of catecholamines, sympathetic activity is increased, whereas parasympathetic activity is decreased, resulting in an increase in blood pressure. HPA, hypothalamic–pituitary–adrenal axis; PVN, periventricular nucleus of the hypothalamus; NMDA, N-methyl-D-aspartic acid.

**Figure 4 ijms-25-13284-f004:**
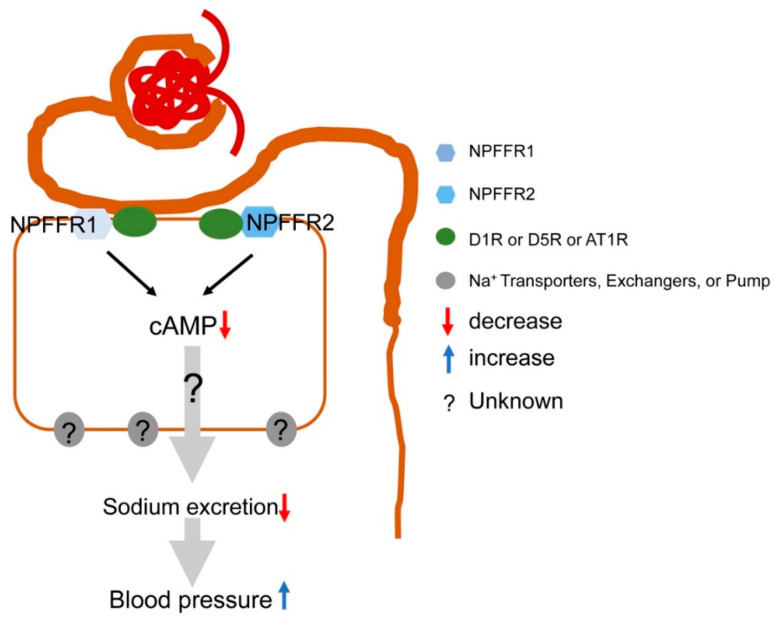
NPFF-mediated regulation of blood pressure in the kidney. The genes and proteins of NPFF and its receptors, NPFFR1 and NPFFR2, are expressed in human and mouse kidneys. NPFF decreases forskolin-stimulated cAMP production. The renal subcapsular infusion of NPFF in C57BL/6 mice decreases renal sodium excretion and increases blood pressure. These findings suggest that NPFF and its receptors in the kidney increase renal sodium transport and subsequently blood pressure, but the mechanisms involved remain to be determined.

**Table 1 ijms-25-13284-t001:** Summary of the neuropeptide FF (NPFF) system [16,17,18,19,20].

**Precursor**		**Pro-NPFF_A_**	**Pro-NPFF_B_**
	Peptide	NPFF	RFRP-1 (NPSF)
		Human: SQAFLFQPQRF	Human: MPHSFANLPLRF
		Mouse: SPAFLFQPQRF	Mouse: VPHSAANLPLRF
		Rat: NPAFLFQPQRF	Rat: VPHSAANLPLRF
		NPAF	RFRP-3 (NPVF)
		Human: AGEGLNSQFWSLAAPQRF	Human: VPNLPQRF
		Mouse: QFWSLAAPQRF	Mouse: VNMEAGTRSHFPSLPQRF
		Rat: EFWSLAAPQRF	Rat: ANMEAGTMSHFPSLPQRF
**Receptor**		**NPFFR2**	**NPFFR1**
	Other names	GPR74, HLWAR77	GPR147, OT7T022
	Cognate peptide	NPFF NPAF	RFRP-1 RFRP-3
	Affinity to NPFF	High	High
	Major signaling	Gαi	Gαi

**Table 2 ijms-25-13284-t002:** Summary of mammalian RFamide peptides [14,21,22,23,24,25,26].

Group	Peptides	Cognate Receptor	Full Name or Alternative Name	G Protein
NPFF	NPFF NPAF	GPR74 (NPFFR2, HLWAR77)	Neuropeptide FF PQRF amide peptide	Gαi/o
GnIH	RFRP-1 RFRP-3	GPR147 (NPFFR1, OT7T022, GnIH receptor)	Gonadotropin-inhibitory hormone RFamide-related peptide	Gαi/o
QRFP	43RFa	GPR103	Pyroglutamylated RFamide peptide	Gαq
	26RFa	(QRFPR)	26RFamide peptide	Gαi
KiSS	Kisspeptin-1	GPR54 (KiSS-1R, hOT7T175, AXOR12)	Kisspeptin	Gαq/11
	Kisspeptin-10			
	Kisspeptin-13			
	Kisspeptin-14			
	Kisspeptin-54			
	Kisspeptin-145			
PrRP	PrRP31 PrRP20	GPR10 (PRLHR, GR3)	Prolactin-releasing peptide	Gαi/o

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
