# Peer review of "An Overview on Renal and Central Regulation of Blood Pressure by Neuropeptide FF and Its Receptors"

_ijms, 2024, doi:10.3390/ijms252413284_

Round 1
Reviewer 1 Report
Comments and Suggestions for Authors
Thank you for giving me the opportunity to review this manuscript. This is an interesting and relevant work that reviews the role of the NPFF system in the regulation of blood pressure focusing on the CNS and kidney. I especially appreciate the figures. One suggestion, that would require a greater effort, is to add another figure showing the data on the CNS expression of NPFF and its receptors.
A few minor details need to be addressed:
1. The abstract is a little bit misleading. In this review, except for the kidney, no other peripheral organs are mentioned. I would suggest leaving out the mention of peripheral organs and just stating kidney (Line 20).
2. Please, revise the abbreviations in the text. In the NPFF and NPFF receptors section, the authors use different abbreviations for NPFF receptors (NPFF-R1, NPFFR1) and the precursors.
3. The writing style of mRNA needs also to be addressed: In line 78 capital letters are used for rats, whereas in line 101 small letters but italic. Please, use the correct form.
4. Please rephrase the sentence in line 120: "NPFFR1 and NPFFR2, as all seven transmembrane spanning receptors...."
5. RF9 is mentioned throughout the review as an NPFFR antagonist. However, it has been implicated that RF9 is a KISS1R agonist and the kisspeptin system has also been implicated in blood pressure regulation and kidney function. Please, address this or at least mention it in the article.
6. Figure 1: As I see it, bolder/thicker arrows indicate the principal mechanisms of NPFFR1 and NPFFR2 activation. Please, indicate it in the caption below.
7. Line 291-296. This paragraph seemed a little confusing. I don't understand the mention of the hypothalamus here, it was already extensively discussed in the previous section. I guess the authors want to address limbic system in this paragraph. I think rephrasing this section would benefit the manuscript.
8. Figure 2: Correction in the caption is needed (line 314) before "adrenergic"
9. Line 351: Please add "dopamine" before D1 receptors
10. Line 375: Correct C57BL/6
Regards,
Author Response
Thank you for giving me the opportunity to review this manuscript. This is an interesting and relevant work that reviews the role of the NPFF system in the regulation of blood pressure focusing on the CNS and kidney. I especially appreciate the figures. One suggestion, that would require a greater effort, is to add another figure showing the data on the CNS expression of NPFF and its receptors.
Response: We appreciate the reviewer’s positive comments. Per the reviewer’s suggestion, a new figure (new Figure 1) is added in Page 4 (The description of NPFF expression, please see the text, Page 3, Line 86-92). The original figures are renumbered accordingly in the revised manuscript. Considering the species variation in NPFFR1 and NPFFR2 expression, Figure 1 only shows NPFF, not NPFFR1 and NPFFR2. However, figure 1 legend (Page 4, Line 117-130) includes a description of the expression of NPFFR1 and NPFFR2 in mice, based on previous report (Lee et al, Sci Rep. 2024;14: 15407. doi: 10.1038/s41598-024-64484-9).
All revisions are highlighted in yellow.
A few minor details need to be addressed:
- The abstract is a little bit misleading. In this review, except for the kidney, no other peripheral organs are mentioned. I would suggest leaving out the mention of peripheral organs and just stating kidney (Line 20).
Response: The words “peripheral organs, especially” is removed in the revised manuscript, which is not able to be shown in Page 1, Line 21.
- Please, revise the abbreviations in the text. In the NPFF and NPFF receptors section, the authors use different abbreviations for NPFF receptors (NPFF-R1, NPFFR1) and the precursors.
Response: The “NPFF-R1” and “NPFF-R2”, Page 2, Line 62, have been changed to “NPFFR1” and “NPFFR2”, respectively. The abbreviations “NPFFR1” and “NPFFR2” are now used throughout the revised version of this manuscript.
- The writing style of mRNA needs also to be addressed: In line 78 capital letters are used for rats, whereas in line 101 small letters but italic. Please, use the correct form.
Response: For rodent gene, only the first letter is capitalized. For human genes, all the letters are capitalized. As the reviewer knows all genes should be italicized.
- Please rephrase the sentence in line 120: "NPFFR1 and NPFFR2, as all seven transmembrane spanning receptors...."
Response: The sentence has been changed to “NPFFR1 and NPFFR2, as with all seven transmembrane spanning receptors, act via heterotrimeric guanine nucleotide regulatory proteins (G proteins).” in the revised version of this manuscript in Page 5, Line 160.
- RF9 is mentioned throughout the review as an NPFFR antagonist. However, it has been implicated that RF9 is a KISS1R agonist and the kisspeptin system has also been implicated in blood pressure regulation and kidney function. Please, address this or at least mention it in the article.
Response: We have included a discussion on this matter in Page 10, Lines 409-414.
- Figure 1: As I see it, bolder/thicker arrows indicate the principal mechanisms of NPFFR1 and NPFFR2 activation. Please, indicate it in the caption below.
Response: The revised caption now includes “The thick arrows show the principal mechanisms of NPFFR1 and NPFFR2 activation while the thin arrows show ancillary mechanisms of NPFFR2 activation”, Page 6, lines 242-243.
- Line 291-296. This paragraph seemed a little confusing. I don't understand the mention of the hypothalamus here, it was already extensively discussed in the previous section. I guess the authors want to address limbic system in this paragraph. I think rephrasing this section would benefit the manuscript.
Response: We agree that statements in that paragraph may be confusing; we have revised the paragraph (Page 8, Lines 333-337).
- Figure 2: Correction in the caption is needed (line 314) before "adrenergic."
Response: We have added a hyphen as in “α- and β-adrenergic” in the new Figure 3, Page 9, Line 355.
- Line 351: Please add "dopamine" before D1 receptors
Response: Per the reviewer’s suggestion, the word “dopamine” is added before “D1 receptors” in Page 10, Line 388, in the revised version of this manuscript.
- Line 375: Correct C57BL/6
Response: Per the reviewer’s suggestion, C57BL/6” is used rather “C57 Bl/6”, Page 11, Line 446.
Reviewer 2 Report
Comments and Suggestions for Authors
In the manuscript (ID: ijms-3311126), authors reviewed the studied the regulation of neuropeptide-2 peptide FF and its receptors on renal and central blood pressure. In general, the content of this manuscript meets the requirements of International Journal of mechanical Sciences. Therefore, I think this manuscript is suitable for publication in International Journal of mechanical Sciences after a major revision.
(1) There are too many proper nouns or abbreviations (such as NPFF, cAMP, NPFFR1, NPFFR2, CNS, GWAS, PVN, NTS, SON, GPR147, OT7TO22, GPR74, HLWAR77, NPVF, HEK293, COS-7, CHO, SH-SY5Y, PrRP, etc.) in the manuscript. The author is advised to add an abbreviation section at the end of the manuscript.
(2) Title: It is suggested that the author revise the title of the manuscript, which is more like a research paper than a review article.
(3) Line 11-13: Neuropeptide FF (NPFF), an endogenous octapeptide originally isolated from the bovine brain, belongs to the RFamide family of peptides that has a wide range of physiological functions and pathophysiological effects. The meaning of the sentence is not clear. It is suggested that the author modify the sentence to make the meaning more clear. In addition, It is recommended to avoid such long sentence.
(4) Line 17: Please provide the full name of cAMP. When an abbreviation appears in the manuscript, write its full name first, and the abbreviation is written after the full name in parentheses. Subsequently, use the abbreviation consistently and do not write out the full term again.
(5) Line 22: Should be “Neuropeptide FF (NPFF),” rather “NPFF”.
(6) Line 22: Should be “NPFF receptor 1 (NPFFR1)” and “NPFF receptor 2 (NPFFR2)” rather “NPFFR1” and “NPFFR2”.
(7) 1. Introduction: This review lacks the description of the writing method, such as which keywords the author chose to consult the literatures, which databases were used, the number of literatures collected and the number of literatures finally selected, etc. Suggest authors to add the method.
(8) Line 51: Please provide the full name of NPVF. When an abbreviation appears in the manuscript, write its full name first, and the abbreviation is written after the full name in parentheses. Subsequently, use the abbreviation consistently and do not write out the full term again.
(9) Line 48-64: Whether the author can use the form of a graph or table to illustrate this part of the content. The present form is not conducive to understanding the content.
(10) 2. NPFF and NPFF receptors and 2. NPFF and NPFF receptors: It is suggested that the author further explain these two parts in the form of graphs, which can increase the readability of the review and help readers understand these contents.
(11) Line 55 and 159: “arginine (R) and an amidated phenylalanine (F)” and “Asp”. It is suggested that the author carefully read the submission requirements of International Journal of mechanical Sciences, and unify the writing of amino acids in this manuscript.
(12) Overall, the authors give a very in-depth review. However, it is difficult for the general reader to understand the review. Therefore, it is suggested that the author try to use the form of graph in the manuscript to make the content more conducive to readers' learning.
Comments on the Quality of English LanguageThe English could be improved to more clearly express the research.
Author Response
(1) There are too many proper nouns or abbreviations (such as NPFF, cAMP, NPFFR1, NPFFR2, CNS, GWAS, PVN, NTS, SON, GPR147, OT7TO22, GPR74, HLWAR77, NPVF, HEK293, COS-7, CHO, SH-SY5Y, PrRP, etc.) in the manuscript. The author is advised to add an abbreviation section at the end of the manuscript.
Response: Per the reviewer’s suggestion, all abbreviations (except HLWAR77 and SH-SY5Y) are now listed in the Abbreviation section at the end of the revised manuscript, Pages 11-12, Line 491-538.
HLWAR77 is NPFFR2 (see Table 2), which was first reported by Elshourbagy NA et al, J Biol. Chem. 2000, 275, 25965-25971 (cited as Ref. 17 in this manuscript). We do not know what the letters of HLWAR stand for in the name of “HLWAR77”.
(2) Title: It is suggested that the author revise the title of the manuscript, which is more like a research paper than a review article.
Response: Per the reviewer’s suggestion, the title of the manuscript is changed to “An overview on renal and central regulation of blood pressure by neuropeptide FF and its receptors”. (Page 1, Lines 2-3).
In the abstract, a sentence “Relevant published articles were searched in PubMed, Google Scholar, Web of Science, and Scopus” is also added. (Page 1, Lines 18-19).
(3) Line 11-13: Neuropeptide FF (NPFF), an endogenous octapeptide originally isolated from the bovine brain, belongs to the RFamide family of peptides that has a wide range of physiological functions and pathophysiological effects. The meaning of the sentence is not clear. It is suggested that the author modifies the sentence to make the meaning clearer. In addition, it is recommended to avoid such long sentences.
Response: the long first sentence in the Abstract is now changed to two sentences, “Neuropeptide FF (NPFF) is an endogenous octapeptide that was originally isolated from the bovine brain. It belongs to the RFamide family of peptides that has a wide range of physiological functions and pathophysiological effects”. (Page 1, Lines 11-13).
(4) Line 17: Please provide the full name of cAMP. When an abbreviation appears in the manuscript, write its full name first, and the abbreviation is written after the full name in parentheses. Subsequently, use the abbreviation consistently and do not write out the full term again.
Response: The full name of cAMP, cyclic adenosine monophosphate, is included in the abstract (Page 1, Line 17).
(5) Line 22: Should be “Neuropeptide FF (NPFF),” rather “NPFF”.
Response: As per the reviewer’s suggestion, “Neuropeptide FF (NPFF)” is included in the Keywords section (Page 1, Line 23).
(6) Line 22: Should be “NPFF receptor 1 (NPFFR1)” and “NPFF receptor 2 (NPFFR2)” rather “NPFFR1” and “NPFFR2”.
Response: As per the reviewer’s suggestion, “NPFF receptor 1 (NPFFR1)” and “NPFF receptor 2 (NPFFR2)” are listed in the Keywords section (Page 1, Line 23).
(7) 1. Introduction: This review lacks the description of the writing method, such as which keywords the author chose to consult the literatures, which databases were used, the number of literatures collected and the number of literatures finally selected, etc. Suggest authors to add the method.
Response: As per the reviewer’s suggestion, a paragraph regarding the literature search is added at the end of “Introduction” section (Page 2, Lines 49-57).
(8) Line 51: Please provide the full name of NPVF. When an abbreviation appears in the manuscript, write its full name first, and the abbreviation is written after the full name in parentheses. Subsequently, use the abbreviation consistently and do not write out the full term again.
Response: Per the reviewer’s suggestion, the abbreviations, such as NPVF are spelled out their first usage (Page 2, Lines 63). There is also an Abbreviations section in the revised manuscript (Pages 11-12, Line 491-538).
(9) Line 48-64: Whether the author can use the form of a graph or table to illustrate this part of the content. The present form is not conducive to understanding the content.
Response: Per the reviewer’s suggestion, two tables, Table 1 (Page 2) and Table 2 (Page 3) are added.
(10) 2. NPFF and NPFF receptors and 2. NPFF and NPFF receptors: It is suggested that the author further explain these two parts in the form of graphs, which can increase the readability of the review and help readers understand these contents.
Response: Reviewer 1 also suggested a graph to illustrate the distribution of NPFF and its receptors, which is now illustrated as Figure 1 (See our response to Reviewer 1). The signaling of NPFF and its receptors were in the original Figure 1 (re-numbered to Figure 2 in this revised version). We hope Figures 1 and 2 can help increase the readability of our revised manuscript.
Figure 1 is in Page 4, revised manuscript.
(11) Line 55 and 159: “arginine (R) and an amidated phenylalanine (F)” and “Asp”. It is suggested that the author carefully read the submission requirements of International Journal of mechanical Sciences and unify the writing of amino acids in this manuscript.
Response We use the full names with one letter symbol in parentheses for amino acids used throughout this manuscript. Thus, “Asp” is changed to “aspartic acid (D)” in this revised version of manuscript, on Page 5, Line 199.
(12) Overall, the authors give a very in-depth review. However, it is difficult for the general reader to understand the review. Therefore, it is suggested that the author try to use the form of graph in the manuscript to make the content more conducive to readers' learning.
Response: We appreciate the reviewer’s overall positive comments. Per the reviewer’s suggestion, we have added Table 1 (page 2), Table 2 (page 3), and a new Figure 1 (Page 4).
Round 2
Reviewer 2 Report
Comments and Suggestions for Authors
The authors have carefully revised the manuscript (ID: ijms-3311126) and the quality of the manuscript has been improved accordingly. Therefore, I think that the manuscript can be accepted for publication in International Journal of Molecular Sciences.